# The Impacts of Quality-Oriented Dataset Labeling on Tree Cover Segmentation Using U-Net: A Case Study in WorldView-3 Imagery

Tao Jiang [1,*], Maximilian Freudenberg [1], Christoph Kleinn [1], Alexander Ecker [2] and Nils Nölke [1]

1   Forest Inventory and Remote Sensing, Faculty of Forest Sciences and Forest Ecology, University of Göttingen, Büsgenweg 5, D-37077 Göttingen, Germany
2   Institute of Computer Science and Campus Institute Data Science, University of Göttingen, Goldschmidtstraße 1, D-37077 Göttingen, Germany
*   Correspondence: tjiang@gwdg.de

**Abstract:** Deep learning has emerged as a prominent technique for extracting vegetation information from high-resolution satellite imagery. However, less attention has been paid to the quality of dataset labeling as compared to research into networks and models, despite data quality consistently having a high impact on final accuracies. In this work, we trained a U-Net model for tree cover segmentation in 30 cm WorldView-3 imagery and assessed the impact of training data quality on segmentation accuracy. We produced two reference tree cover masks of different qualities by labeling images accurately or roughly and trained the model on a combination of both, with varying proportions. Our results show that models trained with accurately delineated masks achieved higher accuracy (88.06%) than models trained on masks that were only roughly delineated (81.13%). When combining the accurately and roughly delineated masks at varying proportions, we found that the segmentation accuracy increased with the proportion of accurately delineated masks. Furthermore, we applied semisupervised active learning techniques to identify an efficient strategy for selecting images for labeling. This showed that semisupervised active learning saved nearly 50% of the labeling cost when applied to accurate masks, while maintaining high accuracy (88.07%). Our study suggests that accurate mask delineation and semisupervised active learning are essential for efficiently generating training datasets in the context of tree cover segmentation from high-resolution satellite imagery.

**Keywords:** dataset quality; semisupervised active learning; U-Net; tree cover segmentation



## 1. Introduction

Trees, as individuals and forests, have varying economic, environmental, ecological and social functions [1,2]. Thus, monitoring them through different sources of data to support forest-related decision making is important. With the advantages of sensor technology and computational power, remote sensing has become an essential data source in forest monitoring. Very high-resolution imagery (e.g., WorldView-3, 30 cm) has proven to be capable of identifying and accurately delineating irregularly and complex-shaped tree cover with submeter spatial resolution [3,4].

Deep learning has emerged as a powerful method in remote sensing imagery analysis due to its high-level feature representations and generalization capabilities [5,6]. Among various deep learning models, semantic segmentation neural networks have been applied successfully to monitoring tree cover from remote sensing imagery. Semantic segmentation neural networks output a pixel-level mask that is the same size as the input image. They assign a class to each pixel within the input image. Fully convolutional neural networks (FCNs) [7] are the backbone of these models, which exclude fully connected layers in the network architecture and ensure the output is an image-like label mask. The U-Net architecture [8] has achieved excellent results on biomedical and other image segmentation

problems by using an encoder?decoder structure that integrates different levels of semantic information to achieve high-resolution segmentation masks. The U-Net architecture has been employed successfully in land use/land cover (LULC) classification [9,10], road extraction [11,12] and change detection [13], to name just a few recent examples from the remote sensing community.

In recent years, there has been growing interest in exploring the potential of U-Net on tree cover and tree crown extraction. Brandt [14] employed U-Net on submeter-resolution imagery to map tree crowns in the vast West African Sahara area, detecting over 1.8 billion individual trees and showing its impressive ability to monitor trees outside of forests across large areas. Mugabowindekwe [15] produced a nation-wide aboveground carbon stock map at tree level by accurately extracting individual tree crowns using U-Net in different landscapes. Freudenberg [16] developed a network that consists of two U-Nets for predicting the tree cover mask and distance transformation, respectively, presenting a strong capability for individual tree crown extraction, both on satellite imagery and aerial images. U-Net has been shown to reach better performance and speed compared to regular CNN models [17,18] when counting oil palm trees in high-resolution imagery [19]. Wagner [20] used U-Net to classify forest types and detect tree crowns and further demonstrated U-Net's applicability for disturbance mapping. Furthermore, variants such as the Residual U-Net and the Attention U-Net have been designed by modifying ResNet [21] and incorporating an attention mechanism [22]. Those U-Net variants have been successfully employed in deforestation monitoring [23,24], individual plant detection [25], tree species classification [26,27] and urban green space extraction [28].

When it comes to dataset labeling, quantity and quality are the two main factors determining success or failure of a given task. Model performance correlates with the quantity of training data (i.e., sample size), and several studies, for example [29–31], have been carried out to explore how the sample size and sample distribution affect the classification accuracy of models. A decline in model performance can be seen in recent studies [25,26,32] through ablation tests, i.e., reducing the sample size gradually. However, whether and how the quality of the dataset affect model performance is still poorly understood. For example, it is still unclear how accurately images should be labeled and how to best select images for labeling from a large number of unlabeled candidate images. Therefore, we investigated the quality of the dataset from two perspectives: image delineation and image selection.

There are many factors influencing the delineation quality; it can vary from person to person and with the time invested. More specifically, unlike buildings or other types of land cover with regular shapes, trees often have complex and irregular shapes, which makes delineation demanding and time consuming with more uncertainty involved. Therefore, it is crucial to know how large the added value of spending more time on delineating is or whether it pays off or not to delineate thoroughly. While there is no general answer to this question, we can attempt to provide guidance for the task of tree cover segmentation. Apart from delineation quality, overall dataset quality is also determined by the selection of the 'most useful and valuable' images to label. This is where active learning comes into play. Active learning is a training strategy that uses an interactive approach to data labeling [33]. It can help to reduce labeling costs significantly and improves performance in object detection [34], image scene classification [35] and biomedical image segmentation [36]. Instead of labeling a large amount of data and training a neural network once, active learning starts with a small amount of training data. The model is then trained with this small amount of data and makes predictions about new data. Data, where the model prediction has a high uncertainty, are labeled by hand and added to the training set. The whole training workflow then comprises several iterations of training, prediction and labeling. In forest monitoring, active learning has been used to estimate oil palm density at the country scale by selecting the most representative samples to be labeled [37]. To further reduce the labeling cost, semisupervised learning [38], which incorporates pseudo labels

(i.e., model predictions) into the training dataset, was shown to increase performance by using unlabeled data.

The overarching goal of this paper was to increase the cost efficiency of dataset labeling in the context of tree cover segmentation from high-resolution imagery. In this case study, we examined three research questions: (1) how can semisupervised active learning help to reduce the effort needed for labeling?; (2) how does accuracy correlate with entropy when employing semisupervised active learning?; (3) how does mask delineation quality affect performance during segmentation?

## 2. Materials and Methods

### 2.1. Study Area and Data Source

Bengaluru (12°58′N, 77°35′E), the capital of the Indian State of Karnataka, is situated on Southern India's Deccan Plateau at an altitude of about 920 m above sea level [39]. Bengaluru used to be known as the 'garden city' of India owing to its widespread parks, green spaces and many alleys with old, towering trees. Even though many of these natural areas and elements have been lost due to infrastructure development, trees and green spaces remain abundant compared to other megacities [40]. Trees in Bengaluru can be found in diverse scenes: streets, forest-like city parks, plantations and fields near farms. This makes Bengaluru a great case for the study of tree cover segmentation.

The Worldview-3 satellite image, acquired on 16 November 2016, has one panchromatic band with a 0.3 m spatial resolution and eight multi-spectral (MS) bands with 1.24 m spatial resolution. In the pre-processing phase, pansharpening was performed using PCI Geomatica to obtain a higher spatial resolution (0.3 m) for the MS imagery [41]. The first MS band (coastal blue) was removed for its high sensitivity to water vapor. As a result, seven bands (Table 1) of WorldView-3 were used for further processing in this research.

A 50 km × 5 km rectangular research transect (Figure 1) in the northern part of Bengaluru was defined in the framework of a larger Indian–German collaborative research project. This transect covers a wide range of areas from densely built-up urban environments to broad rural areas.

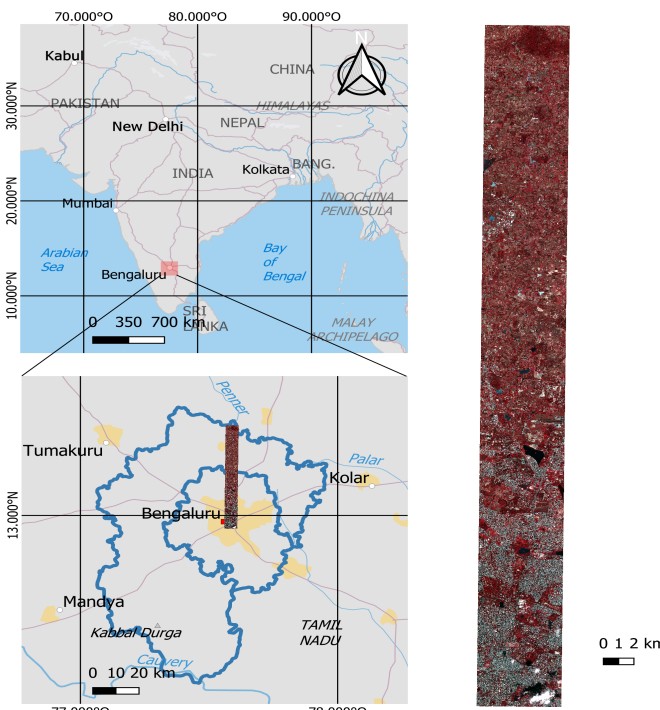

**Figure 1.** Location of the study area: a transect of 50 km × 5 km in the northern part of Bengaluru, India. The transect is enlarged here as a WorldView-3 false color composite (i.e., near IR1, red and green).

**Table 1.** The bands of WorldView-3 imagery used in the research.

| Spectral Band | Wavelength | Spatial Resolution after Pansharpening |
|---|---|---|
| Blue | 450–510 nm | 30 cm |
| Green | 510–580 nm | 30 cm |
| Yellow | 585–625 nm | 30 cm |
| Red | 630–690 nm | 30 cm |
| Red Edge | 705–745 nm | 30 cm |
| Near IR1 | 770–895 nm | 30 cm |
| Near IR2 | 860–1040 nm | 30 cm |

### 2.2. Dataset Delineation Quality

From the preprocessed imagery, 330 image patches with a size of 100 m × 100 m were collected on a systematic grid. Within those image patches, we manually delineated the tree crowns with the two different labeling qualities, as shown in Figure 2. The final dataset was split into training, validation and test sets, containing 280, 20 and 30 image patches, respectively. The images within the test set remained fixed throughout all experiments, while the composition of the training and validation sets varied from experiment to experiment and during cross-validation runs. The two labeling qualities are defined as follows.

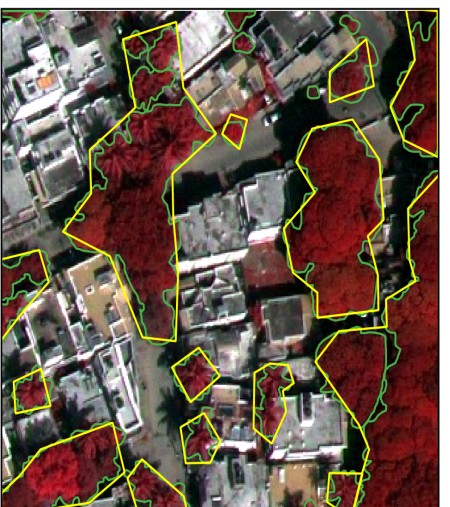 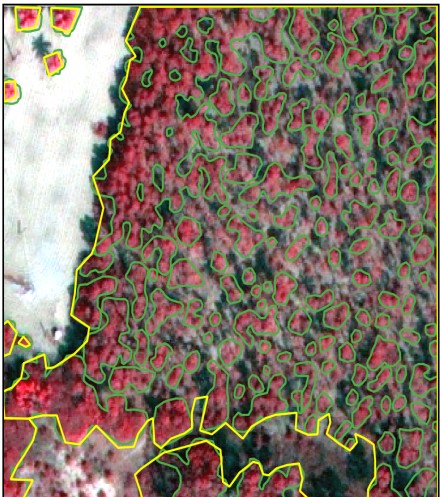

**Figure 2.** Illustration of the two different qualities of delineating tree crowns: an urban (**left**) and a rural (**right**) environment. Yellow denotes rough delineations and green denotes accurate delineations of tree crowns.

Roughly delineated mask: To generate this mask, we labeled tree cover pixels in the image patch as quickly as possible. This means that tree cover was delineated with simple polygons that included small parts of other land cover (buildings, roads or other vegetation) and some omitted tree cover pixels, while still covering the most significant parts of the tree crowns.

Accurately delineated mask: The principle of this labeling strategy was to mask tree cover pixels as carefully as possible. This means that when delineating tree cover, we zoomed in on the boundary between the tree cover and the other land cover types to make sure that we did not include false positives or omit any tree cover pixels. The resulting polygons were more complex and irregular than those obtained by rough delineation (Figure 2).

### 2.3. The U-Net Model

In all experiments, we employed a modified version of U-Net [8]. The modifications we made were concerned with the double-convolutions: We bypassed each double convolution with a 1x1 convolution, whose result was added to the output of the second convolution. Furthermore, we only used four pooling stages, not five as in the original implementation. To improve the resolution of fine-grained details in the prediction [42], skip connections via concatenation were also applied (Figure 3).

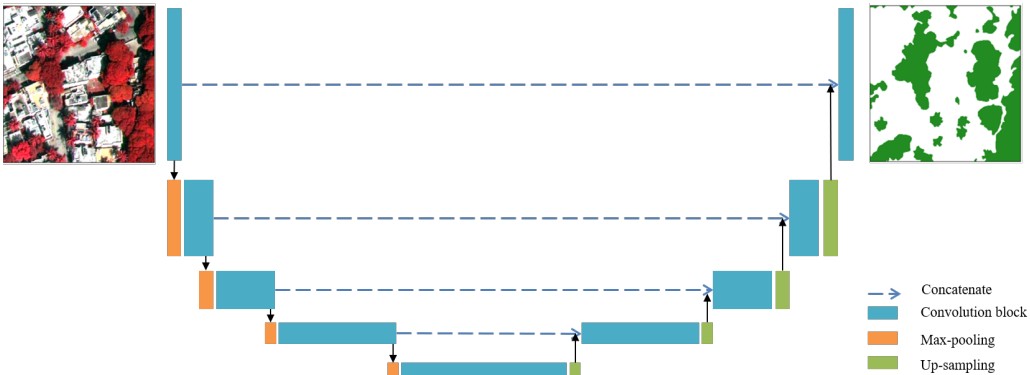

**Figure 3.** The architecture of U-Net.

Data augmentation includes random rotations (90 degrees), vertical and horizontal flipping, as well as random crops to $256 \times 256$ pixels. During training, 280 pairs of image patches and masks were passed to our model, and an Adam optimizer with an initial learning rate of 0.0001 and a cosine decay strategy was adopted to minimize Dice loss and optimize model performance.

### 2.4. Semisupervised Active Learning for Image Segmentation

Semisupervised active learning has already been successfully applied in medical image segmentation [43,44] under varying manners. The core concept of semisupervised active learning is selecting images with high uncertainty for manual labeling and using confident predictions as ground truth (i.e., pseudo labels) to reduce the labeling cost. Here, we designed a semisupervised active learning strategy, which was aimed at tree cover segmentation using U-Net. As illustrated (Figure 4), our semisupervised active learning (SSAL) strategy consists of four steps:

1. Label a portion of image patches (we started at 40) to form an initial training dataset. Train an *initial model* with this dataset.
2. Use this model to make predictions for the remaining, unlabeled image patches (we made predictions for $280 - 40 = 240$ patches). Run the uncertainty analysis explained below (1) for the predictions. Sort them according to their uncertainty.
3. Manually label the N image patches whose predictions have the highest uncertainty, and accept the M predictions with lowest uncertainty as model-labeled masks. N and M are determined by a *percentage* and should sum up to the chunk size that was used to increase the dataset (we had a chunk size of 40).
4. Incorporate human and model-labeled masks into the (initial) training dataset, and train a new model. Then, go back to step 2 and repeat the process until all data are labeled (we processed the remaining 240 images in chunks of 40, resulting in 6 iterations).

In our semisupervised active learning strategy, we employed Monte Carlo dropout [45,46] during training and inference for two reasons: First, the initial model was trained on only a few image patches and masks, which can cause overfitting. Dropout, as a regularization technique, was introduced to alleviate model overfitting. Second, in the model inference phase, Monte Carlo dropout was used to obtain multiple predictions to

evaluate the model's uncertainty. Due to the inherent randomness of dropout, the model output slightly differed for each evaluation. In physics, entropy is a measure for disorder, randomness and uncertainty in a system. In machine learning, particularly in active learning, entropy quantifies the uncertainty of model predictions. Equation (1) defines the entropy:

$$E(y) = -\frac{1}{T}\sum_{t=1}^{T}\frac{1}{X}\sum_{x=1}^{X}(y_{x,t}\log_2 y_{x,t} + (1-y_{x,t})\log_2(1-y_{x,t})). \qquad (1)$$

Here, $y_{x,t}$ is the class probability (1: tree; 0: no tree), $x$ is the pixel index and t indexes the different evaluations during inference, which slightly differed due to dropout. We applied the model $T = 5$ times during inference.

The entropy was calculated on a per-pixel level and then averaged across all pixels. For example, when the model output a probability of 50% for a single pixel being a tree cover, the corresponding entropy was equal to 1, which was also the maximum value. In further analysis, we used this metric to sort the model predictions from low to high uncertainty according to the entropy value. To validate the rationale and feasibility of using the entropy for uncertainty analysis in the context of tree cover segmentation, the relationship between entropy and accuracy in predictions was analyzed: we plotted and visually interpreted the relationship at each step of the semisupervised active learning process. As we had a fully annotated dataset at hand, it was possible to determine the achieved accuracies on model-labeled data.

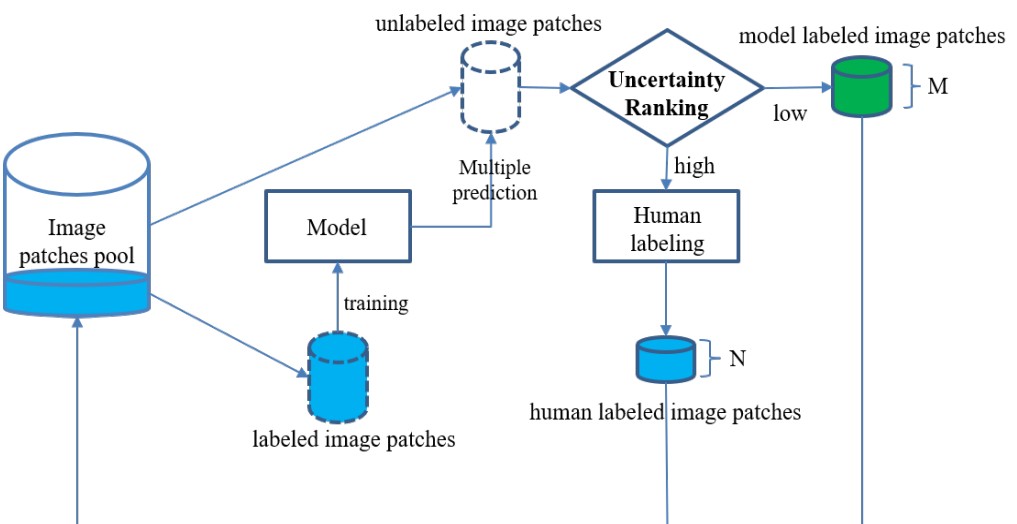

**Figure 4.** The overall flowchart of our *semisupervised* active learning strategy.

### 2.5. Metrics

When comparing model performance, we not only compared the accuracy of the test dataset but also the labeling cost. We recorded the number of roughly and accurately delineated masks we used in different scenarios and then derived the overall time consumption for each dataset. When using the semisupervised active learning strategy, we set the time cost of model-labeled masks to zero. In all scenarios, we kept the number of image patches constant at 280 to eliminate the impact of quantity on model performance.

We used intersection over union (*IoU*) as an accuracy metric, which is also known as the Jaccard index and is described in Formula (2), and the Dice loss function for network backpropagation as described in Formula (3).

$$IoU = \frac{TP}{(TP+FP+FN)} = \frac{\sum(I_t * I_p)}{\sum(I_t + I_p) - \sum(I_t * I_p)}, \qquad (2)$$

$$Loss_{dice} = 1 - \frac{2 * TP}{2 * TP + FP + FN} = 1 - \frac{2 * \sum(I_t * I_p)}{\sum(I_t + I_p)}, \tag{3}$$

where $TP$ are the true positives, $FP$ are the false positives, $FN$ are the false negatives, $I_t$ is the truth label of image and $I_p$ is the model prediction label of image.

### 2.6. An Overview of Different Labeling Strategies

Semisupervised active learning: In this experiment, we compared the data efficiency of three different labeling strategies, random selection, active learning and *semisupervised* active learning, with different percentages of model-labeled masks. For each of the labeling strategies, we started off from an *initial model* trained on 40 randomly selected images. Then, more training data were added in batches of 40 images whose composition was determined by the strategy, until reaching a total of 280 training samples. The random selection corresponded to adding more data to the training dataset without knowing what would increase the model performance most. In the (semisupervised) active learning strategy, we proceeded according to the instructions presented in Section 2.4, and we tested by accepting 0–100% of the model predictions as ground truth. Between 20% and 100%, we took 20% of the steps, resulting in five semisupervised active learning scenarios. For example, a share of 20% meant that 20% of the 40 images in a newly labeled batch came from the model-labeled masks with lowest entropy, and the remaining 80% came from the set of manually delineated masks. The case where 0% of the model predictions were accepted corresponded to conventional active learning. Each model training run was conducted three times to be able to analyze the uncertainty.

### 2.7. Comparisons among Datasets with Different Delineation Qualities

The model was trained using the same image patches but with different masks of different qualities, including pure datasets of roughly and accurately delineated masks (i.e., 100%R and 100%P, respectively) and mixed datasets with different proportions of roughly and accurately delineated masks (i.e., 20%P-80%R, 40%P-60%R, 60%P-40%R and 80%P-20%R). The selection of masks was random and without replacement. To reduce random errors, under each mixed dataset, the random masks selection was executed five times and the average of the accuracies from the five analyses was taken as the final accuracy.

## 3. Results

### 3.1. Comparisons between Different Labeling Strategies

To find the most efficient dataset labeling strategy, we compared (1) semisupervised active learning with (2) standard active learning and (3) randomly selecting training images. All three strategies were applied to roughly and accurately delineated masks.

Figure 5 shows that, in all runs, the final model achieved the highest accuracy in tree cover segmentation and models trained with accurately delineated masks always outperformed those trained on rough masks. Given the same labeling cost, semisupervised active learning always had the highest accuracy, followed by active learning and then random selection. In other words, training with semisupervised active learning converged faster than with active learning, and active learning converged faster than a random selection of training samples. Semisupervised active learning needed the least amount of training data to reach a given performance—but an acceptance rate of model predictions as ground truth (i.e., SSAL 100%) that was too high harmed performance.

The labeling cost for different strategies on the accurately delineated dataset is shown in Table 2. As a result of accepting all model predictions as ground truth, semisupervised active learning with a 100% selection strategy had the lowest labeling cost (only 240 min for labeling the initial 40 images) and produced an accuracy of 84.94% (overall *IoU*). When the share of accepted predictions decreased, performance and labeling cost increased. Semisupervised learning with 60% selection nearly achieved the same accuracy (88.07% overall *IoU*) as pure active learning (88.14%) and halved the labeling cost (816 to 1680 min). A share

of less than 60% of accepted model predictions did not yield a significant performance boost, while they did increase the labeling cost considerably. Regarding the roughly delineated masks (Table 2), semisupervised active learning with 100% selection only took 80 min to label the initial 40 images and yielded a slightly lower accuracy (79.71% overall) than others. However, we also found that the accuracy difference for models trained on rough masks was marginal between different strategies, while the labeling cost changed considerably.

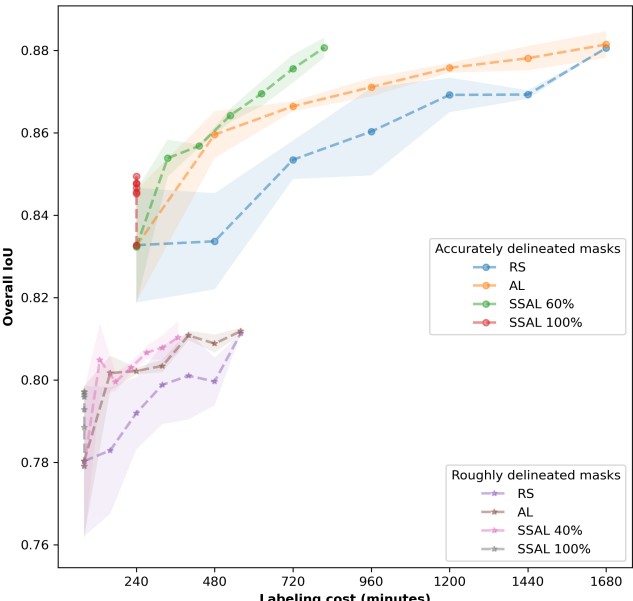

**Figure 5.** Accuracy (overall *IoU*) and labeling cost for different strategies on accurately and roughly delineated masks (i.e., random selection (RS), active learning (AL), semisupervised active learning strategies (SSAL 60%, 40% and 100%)). For accurately delineated masks, the 60% strategy yielded the best results and for roughly delineated masks the 40% strategy yielded the best results.

**Table 2.** Labeling cost and accuracy comparison among active learning (AL) and different semisupervised active learning (SSAL) strategies.

| Strategy | Human-Labeled (A) | Model-Labeled | Labeling Cost | Tree *IoU* | Overall *IoU* |
|---|---|---|---|---|---|
| Accurately delineated dataset | | | | | |
| AL (A) | 280 | 0 | 1680 | 80.33% | 88.14% |
| SSAL (A) 20% | 232 | 48 | 1392 | 80.10% | 88.02% |
| SSAL (A) 40% | 184 | 96 | 1104 | 80.33% | 88.14% |
| SSAL (A) 60% | 136 | 144 | 816 | 80.17% | 88.07% |
| SSAL (A) 80% | 88 | 192 | 528 | 78.73% | 87.06% |
| SSAL (A) 100% | 40 | 240 | 240 | 74.54% | 84.94% |
| Roughly delineated dataset | | | | | |
| AL (R) | 280 | 0 | 560 | 71.67% | 81.19% |
| SSAL (R) 20% | 232 | 48 | 464 | 71.63% | 81.20% |
| SSAL (R) 40% | 184 | 96 | 368 | 71.48% | 81.03% |
| SSAL (R) 60% | 136 | 144 | 272 | 70.82% | 80.52% |
| SSAL (R) 80% | 88 | 192 | 176 | 71.02% | 80.57% |
| SSAL (R) 100% | 40 | 240 | 80 | 69.48% | 79.71% |

Visual inspection of two example images (Figure 6) revealed that tree crown boundaries became more explicit; some small isolated trees were distinguished well in image 150 and small open areas among trees were successfully excluded in image 66 when applying semisupervised active learning with accurately delineated masks. However, applying semisupervised active learning to the roughly delineated dataset failed to improve the segmentation results.

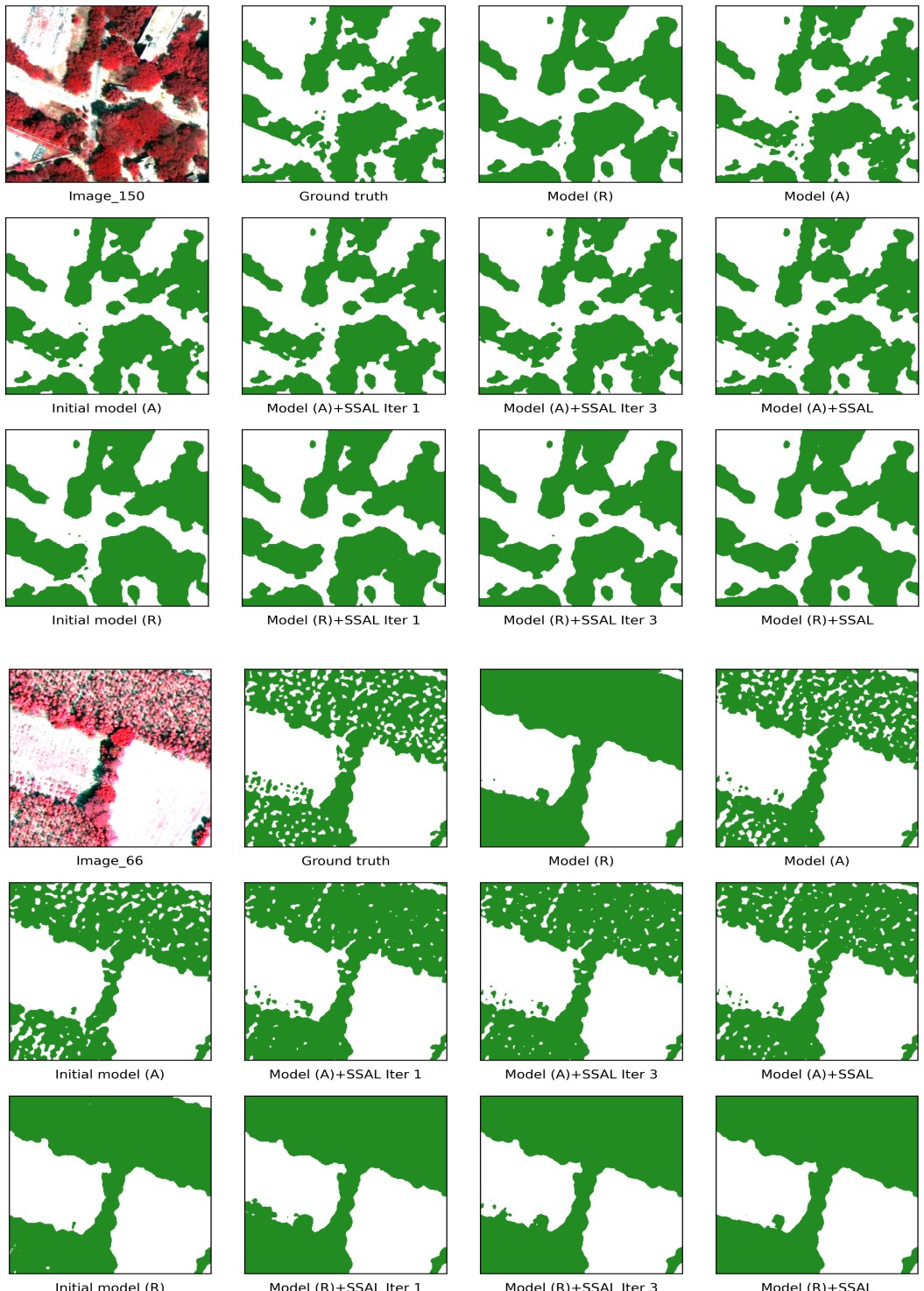

**Figure 6.** Classification results of two test images. For each test image, the subplots in the first row, from left to right, depict the false color image (near IR1, red and green), the ground truth, the segmentation results of models trained on roughly (R) and accurately (A) delineated masks. The subplots in the second and third row represent the classification results of the initial model, the model with intermediate iterations (Iter1 and Iter3) and the final iteration when applying semisupervised active learning (SSAL) on datasets with accurately and roughly delineated masks, respectively.

### 3.2. The Relationship between Entropy and Model-Predicted Tree Cover Accuracy

During semisupervised active learning, we selected images to be labeled and accepted as model predictions of "ground truth" by ranking the predictions according to their

entropy. For that, it was important to analyze the relationship between entropy and tree cover accuracy: Figure 7 shows a negative relationship between entropy and accuracy—low entropy came with high accuracy. With each iteration, as more training data were added, the entropy range declined. Here, semisupervised active learning with a 60% selection strategy was applied in each iteration; 16 images (on the right of each subplot, covered by shaded red areas) were selected to be labeled manually because of the high entropy of their prediction, and predictions of the 24 images with lowest entropy (on the left of each subplot, covered by shaded blue areas) were accepted as masks. As Table 3 depicts, the accuracy increases with each iteration. After six iterations, the model achieved the highest accuracy (80.17% for trees and 88.07% overall) on the test dataset by using 136 accurately delineated masks and 144 model-labeled masks.

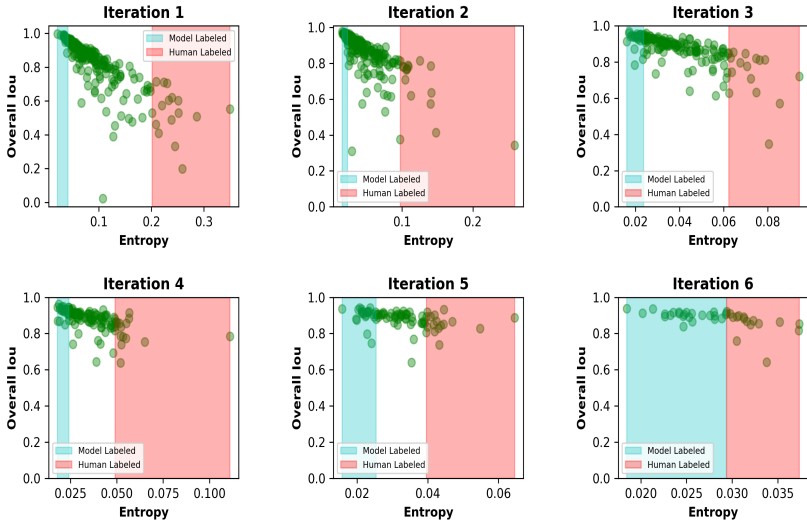

**Figure 7.** The relationship between entropy, accuracy and image mask selection based on semisupervised active learning with a 60% strategy. Each green point represents one image. Points covered by the shaded blue area are accepted model predictions. Points shaded red were selected to be labeled by hand.

**Table 3.** Number of labeled masks and accuracy during semisupervised active learning with the 60% selection strategy.

| Iteration | Human-Labeled | Model-Labeled | Tree *IoU* | Overall *IoU* |
|:---:|:---:|:---:|:---:|:---:|
| 0 | 40 | 0 | 67.15% | 81.41% |
| 1 | 56 | 24 | 76.37% | 85.96% |
| 2 | 72 | 48 | 76.10% | 85.72% |
| 3 | 88 | 72 | 76.77% | 86.20% |
| 4 | 104 | 96 | 78.23% | 87.00% |
| 5 | 120 | 120 | 79.26% | 87.50% |
| 6 | 136 | 144 | 80.17% | 88.07% |

### 3.3. Comparisons among Datasets with Different Delineation Qualities

In the next experiment, we focused on the quality of mask delineation and compared the model accuracies without applying any image selection strategies. Increasing the proportion of accurately delineated masks from 0 to 100% improved the trained model's accuracy (Table 4) from 81.1% to 88.1%. Visual inspection of the results revealed that increasing the proportion of accurately delineated masks improved the level of detail in the predicted tree cover boundary: small and isolated trees were distinguished better and open areas between trees were recognised and excluded (Figure 8).

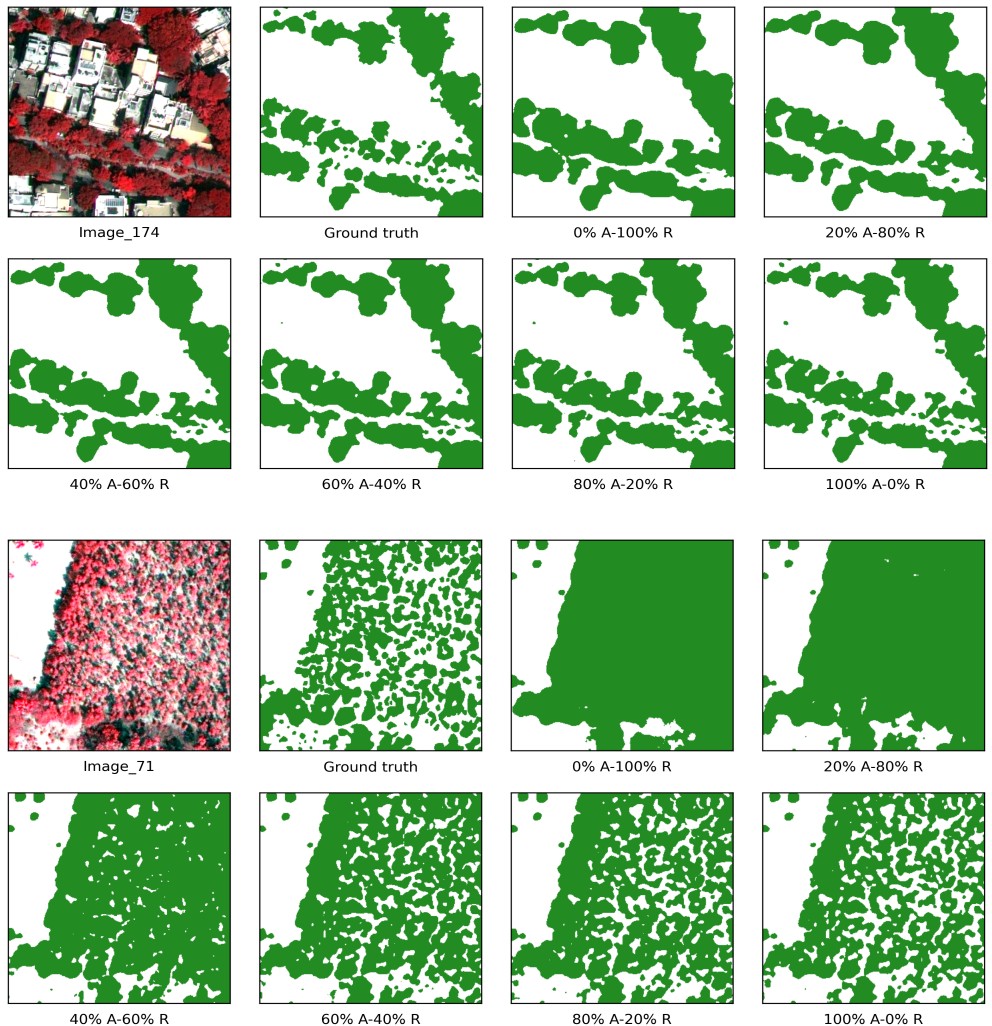

**Figure 8.** Classification results on two test images. In each test image, subplots from left to right then top to bottom represent false color images (near IR1, red, and green), ground truth and classification results from models trained by different proportions of two quality levels of masks (100%R, 20%A, 40%A, 60%A, 80%A and 100%A).

**Table 4.** Labeling cost and accuracy comparison among datasets with different proportions of two quality levels of masks (A: accurately delineated mask; R: roughly delineated mask).

| Datasets | Human-Labeled (A) | Human-Labeled (R) | Labeling Cost | Tree *IoU* | Overall *IoU* |
|---|---|---|---|---|---|
| 100% R | 0 | 280 | 560 | 70.45% | 81.13% |
| 20% A–80% R | 56 | 224 | 784 | 72.78% | 82.69% |
| 40% A–60% R | 112 | 168 | 1008 | 74.85% | 84.43% |
| 60% A–40% R | 168 | 112 | 1232 | 76.14% | 85.62% |
| 80% A–20% R | 224 | 56 | 1456 | 78.24% | 87.01% |
| 100% A | 280 | 0 | 1680 | 79.65% | 88.06% |

We also calculated the labeling cost for every dataset to show how it affected model accuracy by recording the time spent on labeling each accurately or roughly delineated mask (i.e., 6 min and 3 min, on average). The 100% roughly delineated dataset produced the lowest accuracy (81.13%), which was considerably lower than the 100% accurately delineated dataset (88.06%). However, the latter took three times more time to label than the former (Table 4). Similarly, an accuracy improvement (nearly 6%) could be seen by increasing the share of accurately delineated masks from 20% to 80%, while the labeling cost almost doubled (784 to 1456 min).

## 4. Discussion

This research was aimed at giving a better insight into how dataset quality affects the accuracy of automated tree cover segmentation using deep learning models. We interpreted the concept of dataset quality from two aspects: how to select images to be labeled and how to label each image. A semisupervised active learning approach with an accurately delineated mask strategy was found to be an efficient dataset labeling strategy to maintain high segmentation accuracy and save on labeling costs.

We tuned the share of predictions that were accepted as ground truths in our semisupervised active learning strategy to find an optimal value that maintained model accuracy and required a low labeling cost. For the dataset with accurately delineated masks, setting the share to, e.g., 80% or 100% (which translated to accepting (almost) all predictions), resulted in a large drop in labeling costs, but also in segmentation accuracy. This showed that at least a minimum amount of supervision is required to obtain an acceptable performance and that the models could not learn from their outputs alone. In our experiment, the 60% acceptance strategy with accurate masks maintained high accuracy, reduced the labeling cost by more than half and, thereby, proved to be the optimal value. However, it has to be noted that this value was dataset and task dependent and, hence, it could only serve as a rough guidance. It has to be noted that, according to Figure 5, our model's accuracy did not converge; more training data would probably yield even better results. In the case of roughly delineated masks, the accuracy increased only with the first active learning iteration, after that, no significant improvement could be seen. This indicated that the roughly delineated masks were easier to learn and that adding more low-quality training data did not improve the segmentation of finer details.

Regarding how mask quality affects a model's performance, analyses and discussions are scarce. Some studies [25,26,32] have only focused on the impact of sample size on model performance and did not conduct experiments on mask quality. A recent study [31] was conducted to explore how sample labeling affected Mask-RCNN performance on tree crown detection. However, its sample labeling was achieved by deleting entire tree samples from the original dataset to create a different sample distribution, rather than focusing on how accurately to label each tree crown. In some studies [47,48], CNN and Mask-RCNN were found to be less sensitive to delineation accuracy, and detected objects that were overlooked during delineation. However, our results showed that the semantic segmentation model (e.g., U-Net) was significantly affected by delineation quality, resulting in an accuracy disparity between accurately delineated and roughly delineated masks. In our comparison of training datasets with rough and accurate masks, it was shown that the segmentation accuracy increased with an increasing proportion of high-quality masks (by 7 percentage points in total). However, the higher mask quality took three times longer to label. One could argue that it is to be expected that trained models adapt to the respective labeling style, but it could also be the case that models, for example, simply learn to label all green objects as trees, based on their color alone. Consequently, it is now clear that models adapt the labeling style with which they were trained; models trained with roughly delineated masks tended to fill gaps between adjacent trees or omit small trees, while those trained with accurately delineated masks were more accurate in general.

## 5. Conclusions

Utilizing deep convolutional neural networks to segment tree cover in high-resolution imagery remains a challenge, primarily due to time-consuming image labeling work. This study has identified an optimized labeling strategy: utilizing supervised active learning with accurately delineated masks can significantly reduce labeling costs and paves the way for more efficient tree cover segmentation. We believe that this strategy can also be used to tackle deforestation mapping, LULC classification and other tasks. In future work, we plan to investigate how to apply supervised active learning to the image region level to further reduce the labeling effort, advance model uncertainty analysis and explore the potential of more advanced semantic segmentation networks on this topic.

**Author Contributions:** Dataset preparation, experiments, methodology and the first draft were finished by T.J.; M.F. was in charge of the experiment designs and programming guidance; supervision, review and editing were conducted by N.N., A.E. and C.K. All authors have read and agreed to the published version of the manuscript.

**Funding:** This research was funded by the China scholarship council (CSC) and the German Research Foundation (DFG) (Grant Number: KL 894/23-2 and NO 1444/1-2; Research Unit: FOR2432/2).

**Data Availability Statement:** The remote sensing imagery used in this study is subject to a restrictive commercial license and as of the publication date cannot be shared publicly.

**Acknowledgments:** The authors gratefully acknowledge the financial support provided by the China scholarship council, CSC. They also thank the German Research Foundation, DFG, for making the data available. They are also thankful for the cooperation and infrastructural support provided by Indian partners at the Institute of Wood Science and Technology (IWST), Bengaluru.

**Conflicts of Interest:** The authors declare no conflict of interest.

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
