# Peer review of "The Impacts of Quality-Oriented Dataset Labeling on Tree Cover Segmentation Using U-Net: A Case Study in WorldView-3 Imagery"

_remotesensing, doi:10.3390/rs15061691_

Round 1

Reviewer 1 Report

The impacts of quality-oriented dataset labeling on tree cover segmentation using U-Net were studied in this paper. The studied topic is interesting and the paper is generally well written.  Here are the comments.

1) The authors are suggested to explain further why U-Net model is used in this study.

2) The literature review should be well improved with more recently published works reviewed.

3) Technical novelty is limited in this paper. The authors should choose the right article type. A case report or application note is more suitable.

4) Some typos are observed. 

Author Response

Thanks for your comments and suggestions, below are our responses:

1) The authors are suggested to explain further why U-Net model is used in this study.

A: Thanks for this suggestion, we added a brief introduction in paragraph 3 to explain why Unet is appropriate, “The U-Net architecture [8] achieved excellent performance on biomedical and other image segmentation problems by using an encoder-decoder structure that integrates different levels of semantic information to achieve high-resolution segmentation masks.’’ Line 33-35.

Furthermore, we added 5 references of its application in remote sensing field. “The U-Net architecture has been employed successfully in land use/land cover (LULC) classification [9, 10], road extraction [11, 12] and change detection [13], to name just a few recent examples from the remote sensing community.’’. Line 36-38.

Furthermore, 6 more references of Unet’s successful application on this topic--tree cover and tree crown extraction--were added in later part. ‘’Mugabowindekwe [15] produced a nation-wide aboveground carbon stocks map on tree level by accurately extracting individual tree crown using U-Net in different landscapes. Freudenberg [16] developed a network, which consists of two U-Nets for predicting tree cover mask and distance transformation respectively, presenting its strong capability for individual tree crown extraction both on satellite imagery and aerial images. Furthermore, variants such as Residual U-Net and the Attention U-Net have been designed by modifying ResNet [21] and incorporating an attention mechanism [22]. Those U-Net variants have been successfully employed in deforestation monitoring [23, 24], individual plant detection [26, 27], tree species classification and urban green space extraction [28].’’ Line 44-56.

2) The literature review should be well improved with more recently published works reviewed.

A: Thank you for pointing out this issue. We performed another literature search and identified more recently published papers in the context of our topic (Line 33-66). Most of them are from year 2022. They were added, and are:

Five papers on Unet application for remote sensing: [9, 10, 11, 12, 13]

Six papers on Unet application in tree cover and tree crown topic [15, 16, 23, 24, 27, 28]

Four papers on dataset labeling impact on model performance [29, 30, 31, 32]

Two other papers on compute vision [21, 22]

3) Technical novelty is limited in this paper. The authors should choose the right article type. A case report or application note is more suitable.

A: We agree with the reviewer’s first statement but disagree with the conclusion drawn: Coming up with a new architecture or learning method was not the goal of our paper. The main contribution of our paper is an empirical investigation of how one should go about annotating data to efficiently identify tree cover from satellite images. As the process of data annotation is fundamental to any deep learning based solution, the issue of how precisely one should annotate is a crucial question that has not been addressed before in the context of tree cover identification.

4) Some typos are observed.

A: Thank you for pointing this out. We have inspected the entire paper again to eliminate typos.

Reviewer 2 Report

In this paper, the authors do many experiments to prove that datasets with quality labeling be crucial to the final segmentation accuracy, although spend a huge time manually annotating the ground truth. Furthermore, a semi-supervised active learning technique is introduced to reduce manual labeling costs while keeping segmentation accuracy. The paper is well organized, but I think the conclusion is obvious, and the innovation is much limited. What's more, quality labeling is undesired in practical application, compared with rough labeling. My other comments are as follows:

1. Lack of research motivation, such as “the impact of the quality of the dataset has not yet been analyzed in the context of tree cover segmentation”, “it is crucial to know how much time should be invested into labeling at which quality to obtain the desired result” are all less urgent.

2. It will be more valuable when achieving competitive segmentation accuracy with rough labeling. I suggest the authors focus on the uncertainty region in a roughly labeled image but not the image level among the whole datasets.

3. The content in Lines 111-116 overlaps with those in Lines 97-101.

Author Response

Thanks for your comments and suggestions, below are our responses:

  1. Lack of research motivation, such as “the impact of the quality of the dataset has not yet been analyzed in the context of tree cover segmentation”, “it is crucial to know how much time should be invested into labeling at which quality to obtain the desired result” are all less urgent.

A: We are unsure about the reviewer’s rationale here. If getting quality segmentations is not desirable in practice, what then should be the goal of image segmentation? At the end of the day, any image segmentation system will be limited by the amount of data it is trained on and the quality of the labels in the training data – irrespective of what architecture or learning objective is being employed. To what extent one should focus on annotating a large number of images roughly or fewer images precisely is a key question that determines how fast one will achieve a working segmentation system in practice. We believe that this point is spelled out well in the introduction and would appreciate any guidance on what aspect of our reasoning is unclear or, in the reviewer’s opinion, potentially wrong.

To further highlight our motivation, we modified the two places that the reviewer pointed out.

For the first part, we changed lines 62-66:

‘’However, whether and how the quality of the dataset affect model performance is still poorly understood. For example, it is still unclear how accurately images should be labeled and how to best select images for labeling from a large number of unlabeled candidate images. Therefore, we investigated the quality of the dataset from two perspectives: image delineation and image selection.

’’ Line 62-66.

For the second part, we changed lines 71-73 into:

‘’Therefore, it is crucial to know how large the added value of spending more time on delineating is or whether it pays off or not to delineate thoroughly’’ Line 71-73.

  1. It will be more valuable when achieving competitive segmentation accuracy with rough labeling. I suggest the authors focus on the uncertainty region in a roughly labeled image but not the image level among the whole datasets.

A: Thanks for your suggestions. This is an interesting avenue for future research, but orthogonal to our approach in this paper. Here we asked a different question: Using current state-of-the-art methods, how should one go about annotating data? The paper provides a clear answer to this question. We mention in the conclusion that there are of course other approaches to tackle the issue and reduce the labeling effort:

‘’In future work, we plan to investigate how to apply semi-supervised active learning on image region level to further reduce the labeling effort, advance model uncertainty analysis, and explore the potential of more advanced semantic segmentation networks on this topic.’’ Line 337-340

  1. The content in Lines 111-116 overlaps with those in Lines 97-101.

A: Good catch, Lines 111-116 have been deleted to avoid repetition.

Reviewer 3 Report

It is a very interesting work. Surprising that the final accuracy drops so little using a rough delineated mask. Suggest to follow on the research for other topics, such as mapping deforestation and land use change.

Author Response

Thanks for your comments and suggestions, below are our response:

1. It is a very interesting work. Surprising that the final accuracy drops so little using a rough delineated mask. Suggest to follow on the research for other topics, such as mapping deforestation and land use change.

A: Thanks for your comments. The final accuracy difference seems not very large (around 7 percentage points), but it also means that carefully delineating is worth it when you want to achieve better result. Besides, the figure 8 also shows the classification results on both test samples are getting better when increasing the proportion of accurately delineated masks.

For mapping deforestation and land use change topic, it is very good proposal, we decided to include them into conclusion part.

‘’We expect that this strategy can also be used to tackle with deforestation mapping, LULC classification and other tasks.’’ Line 336-337

Reviewer 4 Report

The article is interesting and generally well-writen. The approach focused on data labeling and image segmentation with the use of CNN is interesting and novel. 

However, there are some sections in the article that need to be double checked and corrected.

Some positions in the literature review are missing regarding the subject. It must be added either in introduction or method section. There are 2 main topics to be expendad regarding following:
1) The use of U-net in Remote Sensing in general (there have been many articles published recently in the subject) - at least 5 more papers
2) Research interests on labeling regarding the accuracy of RS data, especially tree crown segmentation and/or classification - at least 3 papers
(ex. https://www.mdpi.com/2072-4292/14/7/1561)

Therefore, some conclusions in section 4 might be corrected, ex. that authors are the first to focus on the impact of mask quality on model performance.

Of course, the approach based on scoring each type of labeling is novel and might be a big help fo researchers at the stage of chosing the specific method at the stage of tree labeling.

Furthermore, the list of bands of WV-3 used in the research could be added in the form of table, while it might not be clear for all readers which bands exactly you have used for the research.

Also, the section 4 should be splitted into 2 - Discussion and conclusion should be in separate sections. Discussion section should focus more on already published papers which focus on labeling regarding model performance in the area of Remote Sensing. I recommend at least 4-5 papers to be compared to your results. As I written above, your approach might be novel, but in this field (with different approaches) there are many research papers in recent years. It could contain, ex. how vectorization affects model quality, how tile size affects model quality, etc.

Additionally, if you could publish your dataset (with all the variants of labeling), that could be very helpful for the researchers in this field.

All in all, I believe that that the paper is valuable for researchers worldwide. However, some corrections, mentioned above should be considered to be applied for the article.

Author Response

Thanks for your comments and suggestions, below are our responses:

1) The use of U-net in Remote Sensing in general (there have been many articles published recently in the subject) - at least 5 more papers

A: Thank you for raising this point. We added 5 references of its application in the remote sensing field to the introduction,

“The U-Net architecture has been employed successfully in land use/land cover (LULC) classification [9, 10], road extraction [11, 12] and change detection [13], to name just a few recent examples from the remote sensing community.’’. Line 36-38.

2) Research interests on labeling regarding the accuracy of RS data, especially tree crown segmentation and/or classification - at least 3 papers

(ex. https://www.mdpi.com/2072-4292/14/7/1561)

Therefore, some conclusions in section 4 might be corrected, ex. that authors are the first to focus on the impact of mask quality on model performance.

A: Thank you for pointing out this study. However, we would like to clarify that it addresses a different question with respect to labelling accuracy: How does not labelling a certain fraction of trees at all affect performance? In our study, the focus is on how rough vs. accurate delineations affect segmentation performance. Thus, the paper is related but does not undermine the novelty of our work. We added a sentence discussing this paper and explaining how it differs from our work to our discussion of related work (Line 314-318):

“A recent study [31] was conducted to explore how the sample labeling affected Mask-RCNN performance on tree crown detection. However, its sample labeling was achieved by deleting entire tree samples from the original dataset to create a different sample distribution rather than focusing on how accurately to label each tree crown.” (Line 314-318)

Furthermore, we added more references regarding sample labeling (sample size and sample distribution), here are our modifications:

‘’Model performance correlates with the quantity of training data (i.e., sample size), and several studies, for example [29], [ 30] and [31], have been carried out to explore how the sample size and sample distribution affect classification accuracy of models. Model performance declines can be seen in recent studies [25 ,26 ,32] through ablation tests, i.e., reducing the sample size gradually.’’ Line 57-62.

We also deleted ‘’ex. that authors are the first to focus on the impact of mask quality on model performance’’.

Of course, the approach based on scoring each type of labeling is novel and might be a big help fo researchers at the stage of chosing the specific method at the stage of tree labeling.

3) Furthermore, the list of bands of WV-3 used in the research could be added in the form of table, while it might not be clear for all readers which bands exactly you have used for the research.

A: Thank you for this good suggestion. We added Table 1, listing the used WorldView-3 bands. Line 110-111.

4) Also, the section 4 should be split into 2 - Discussion and conclusion should be in separate sections. Discussion section should focus more on already published papers which focus on labeling regarding model performance in the area of Remote Sensing. I recommend at least 4-5 papers to be compared to your results. As I written above, your approach might be novel, but in this field (with different approaches) there are many research papers in recent years. It could contain, ex. how vectorization affects model quality, how tile size affects model quality, etc.

A: Thank you for those good suggestions.

Firstly, we have split discussion and conclusion, and added more comparisons with other papers to the discussions part, below are modifications:

‘’Regarding how mask quality affects model performance, analyses and discussions are scarce. Some studies [25 ,26, 32] have only focused on the impact of sample size on model performance but did not conduct experiments on mask quality. A recent study [31] was conducted to explore how the sample labeling affected Mask-RCNN performance on tree crown detection. In fact, its sample labeling was achieved by deleting entire tree samples from the original dataset to create different sample distribution rather than focusing on how accurately to label each tree crown. In some studies [47, 48], CNN and Mask RCNN were found less sensitive to delineation accuracy, thus detecting objects that were overlooked during delineation. However, our results showed that the semantic segmentation model (e.g., U-Net) is significantly affected by delineation quality, resulting in accuracy disparity between accurately delineated and roughly delineated masks.’’ Line 312-322.

Regarding vectorization:

We assume the reviewer meant “rasterization” (the process of converting vector data to raster data) instead of “vectorization”. If not, please let us know. We believe that the rasterization algorithm in use does not significantly affect the resulting dataset, as eventual differences can only occur at object boundaries, which represent only a small fraction of the annotated pixels.

Regarding the tile size:

As fully convolutional neural networks are translation invariant, we expected that the tile size has no big effect on the segmentation quality, as long as it is larger than the typical size of the objects to be detected. The tile size we chose equals approximately 75x75m on ground, which is large enough to capture even the biggest trees in their context, while allowing to perform active learning efficiently.

5) Additionally, if you could publish your dataset (with all the variants of labeling), that could be very helpful for the researchers in this field.

A: Thanks for this suggestion, we will consider to publish our dataset if the paper is accepted. However, the WorldView-3 imagery sadly is subject to a restrictive license.

All in all, I believe that that the paper is valuable for researchers worldwide. However, some corrections, mentioned above should be considered to be applied for the article.

Round 2

Reviewer 2 Report

I do not mean quality image segmentation is not desirable in practice, but mean focusing on quality annotating is less urgent or practical than building a more robust model that is friendly to rough labeling. Or, this paper is more like an experience report but it lacks scientific innovation.